# TCNformer Model for Photovoltaic Power Prediction

**Shipeng Liu [1,2], Dejun Ning [1,\*] and Jue Ma [1,2]**

1    Shanghai Advanced Research Institute, Chinese Academy of Sciences, Shanghai 200120, China
2    University of Chinese Academy of Sciences, Beijing 100049, China
\*    Correspondence: ningdj@sari.ac.cn; Tel.: +86-186-1653-6063

**Abstract:** Despite the growing capabilities of the short-term prediction of photovoltaic power, we still face two challenges to longer time-range predictions: error accumulation and long-term time series feature extraction. In order to improve the longer time range prediction accuracy of photovoltaic power, this paper proposes a seq2seq prediction model TCNformer, which outperforms other state-of-the-art (SOTA) algorithms by introducing variable selection (VS), long- and short-term time series feature extraction (LSTFE), and one-step temporal convolutional network (TCN) decoding. A VS module employs correlation analysis and periodic analysis to separate the time series correlation information, LSTFE extracts multiple time series features from time series data, and one-step TCN decoding realizes generative predictions. We demonstrate here that TCNformer has the lowest mean squared error (MSE), mean absolute error (MAE) and mean absolute percentage error (MAPE) in contrast to the other algorithms in the field of the short-term prediction of photovoltaic power, and furthermore, the effectiveness of each module has been verified through ablation experiments.

**Keywords:** transformer; SkipGRU; TCN; photovoltaic power prediction; time series data prediction





## 1. Introduction

At present, with the rapid development of perovskite solar cell technology [1,2], the maximum efficiency [3] and stability [4] of photovoltaic power have been greatly improved. Photovoltaic power is increasingly important in the field of new energy. According to the data of the International Energy Agency (IEA), the growth rate of global photovoltaic installed capacity has reached as much as 49%. It is estimated that global photovoltaic power will reach 16% of the total power in 2050 [5]. At the same time, China is promoting the construction of a new power system with new energy as the principal part. Photovoltaic power using solar energy is an important branch of new energy and one of the important means for China to achieve the goal of carbon neutrality. After the large-scale integration of photovoltaic power stations into the energy network, the manner by which to accurately predict photovoltaic power and then accordingly dispatch the power grid has become an urgent problem to be addressed. Therefore, improving the prediction accuracy of photovoltaic power is significant for improving the operation efficiency of power stations themselves and for maintaining the stability of power grids.

Many scholars in China and abroad have carried out a lot of research on the prediction of photovoltaic power. At present, the mainstream prediction methods focus on traditional random learning and deep learning methods. In the field of traditional random learning, literature [6] uses historical weather data and historical power data as inputs of a support vector machine (SVM) to build a short-term photovoltaic power prediction model, which has a higher level of accuracy than the traditional autoregressive model (AR) or the radial basis function (RBF) models. One study [7] proposed a model based on Support Vector Regression (SVR) and achieved better prediction performance. In the field of deep learning, recurrent neural network (RNN) structures, such as long short-term memory (LSTM), gated recurrent unit (GRU), and seq2seq structural models, are widely used to analyze and predict time series data for such applications as stock price prediction [8], gold price prediction [9],

traffic flow [10], voice classification [11], etc. The prediction of photovoltaic power can also be regarded as a kind of time series data prediction, so the above algorithms have been used to predict short-term global horizontal irradiance (GHI) or comprehensive solar loads [12,13]. Furthermore, in order to ensure the accuracy as much as possible and reduce the training time, the GRU network has been applied to short-term photovoltaic power prediction [14], and the multivariable GRU model [15–17] has been used to predict solar irradiance or power. Some hybrid models have been applied in the field of photovoltaic power generation prediction, such as the combination of a deep learning model and a heuristic algorithm [18,19], the combination of a deep learning model and a traditional random learning method [20,21], the combination of multiple deep learning models [22,23], etc. The seq2seq structural model represented by the Transformer series model takes the photovoltaic power prediction problem as a experimental sample of its model, such as Autoformer and Informer [24,25]. However, in these models, usually the photovoltaic power prediction data are only used for prediction; that is, the corresponding weather data is not fully used, and the time series features of the data are not fully extracted.

Compared with traditional LSTM, GRU, and other models, the Transformer series seq2seq model can avoid the problem of error accumulation and read longer input data [26], but it is still limited by the length of the input data. It is difficult for the seq2seq series model to capture longer time series features. For this problem, [27] proposes long- and short-time series network (LSTnet) models. The Skip recurrent neural network (Skip RNN) structure is used to capture more long-term time series features.

Based on the above analysis, the current research mainly focuses on the prediction of data within a few hours. When applied to predict a longer time range [28] for photovoltaic power, these methods typically suffer from two major challenges: error accumulation and long-term time series feature extraction in order to simultaneously extract multiple time series features in the historical data of photovoltaic power and weather factors, and to avoid error accumulation. Inspired by the application of LSTM, LSTnet, and Transformer series models in the field of photovoltaic power prediction, this paper proposes a long and short temporary correction network (TCNformer), and we verified the model by using the real data of a photovoltaic station in Australia. According to the experimental results, the TCNformer model greatly optimizes various indicators compared with LSTM, SkipGRU, Transformer, and Informer, improving the accuracy of photovoltaic power prediction.

The contributions of this paper include the following:

(1) According to the different impacts of various weather factors on photovoltaic power generation, a VS module was designed to screen and process data through correlation analysis and periodic analysis of data.
(2) Aiming at the challenge to extract long-term time series features due to the limitation of the traditional Transformer, a LSTFE module was designed to extract multiple time series features through LSTM and a SkipGRU network.
(3) In order to improve the temporal feature extraction and avoid error accumulation, one-step temporal convolutional network (TCN) decoding was used to realize the generative prediction.

## 2. Preliminary

### 2.1. Time Series Features of Photovoltaic Power Data

According to the literature [29,30], the current photovoltaic power prediction problem is usually defined as a time series data prediction problem. However, as the time granularity increases, the degree of the photovoltaic power data affected by external factors increases, and the self-similarity decreases. The basic photovoltaic power data studied in this paper are collected at a 15-min granularity, and they are greatly affected by external factors that have a certain regularity and contingency, so the statistical features of photovoltaic power data show certain periodicity, abruptness, and contingency.

As shown in Figure 1, the 4-day power history data of a photovoltaic station were randomly selected, showing obvious periodicity and volatility.

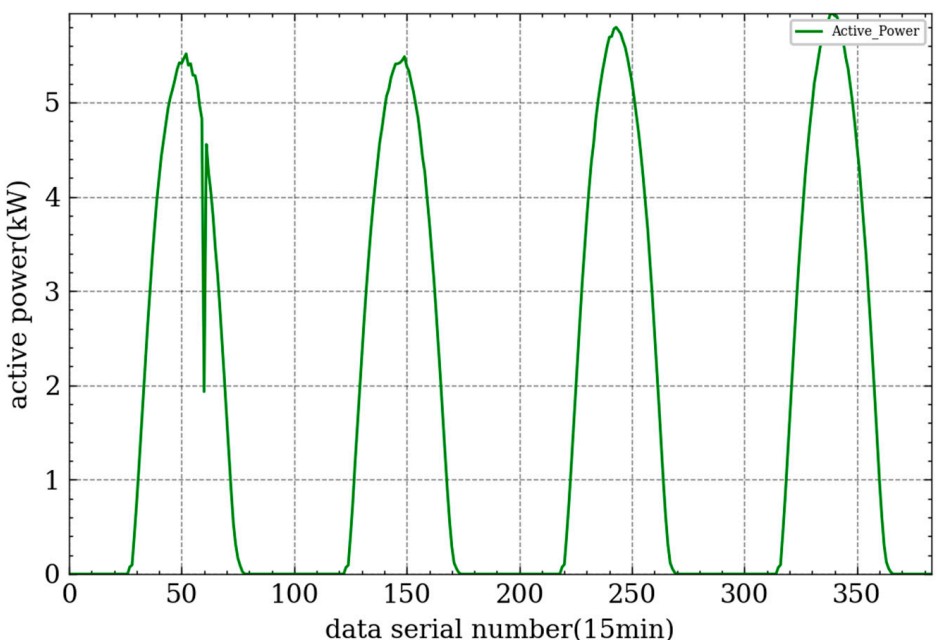

**Figure 1.** Graph of 15-min data from 4 continuous days.

As shown in Figure 2, in order to explore the long-term time series features of photovoltaic power data, this study employed the classic skills of a seasonal prediction model to select the historical data of a photovoltaic power station at 8:30 for 4 consecutive years. Although they show greater volatility, a certain periodicity can still be seen.

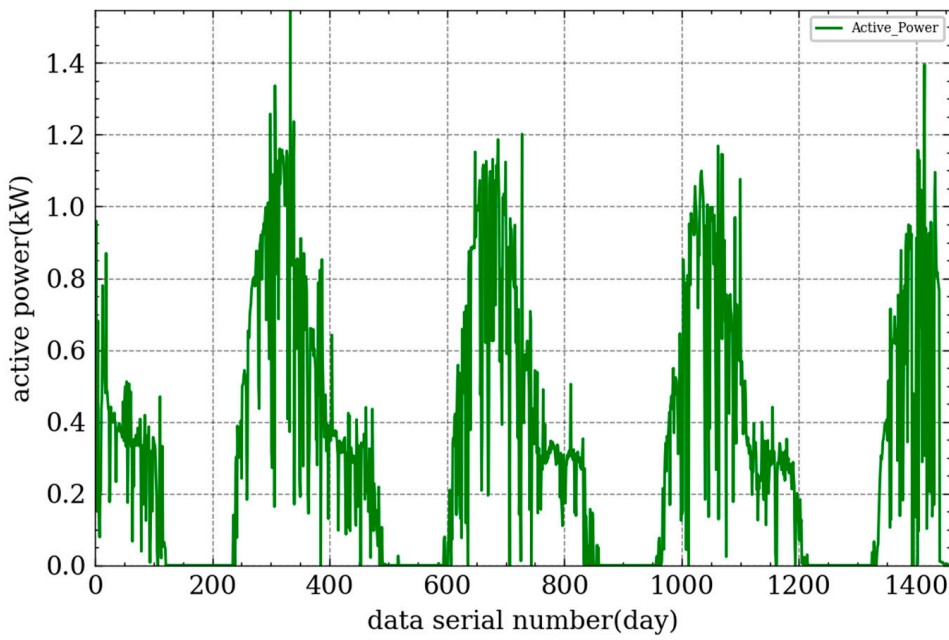

**Figure 2.** Graph of 8:30 data for 4 continuous years.

## 2.2. LSTM and SkipGRU

LSTM is a classic model in the field of time series prediction. In the prediction process, LSTM updates the internal state and the external state at the same time, mainly through three gates: a forgetting gate, an input gate, and an output gate.

The GRU network [31] is a variant of the LSTM network, which combines the three gates of the LSTM unit into two gates. The SkipGRU module skips the connection layer. By

sampling at intervals, it can look back for a longer period of time when the length of the sampling sequence remains unchanged so as to capture the long-term features.

### 2.3. Self-Attention Mechanism and ProbSparse Self-Attention Module

The calculation formula of a traditional self-attention mechanism is as follows:

$$Q, K, V = XW^Q, XW^K, XW^V \tag{1}$$

$$A(Q, K, V) = softmax\left(\frac{QK^T}{\sqrt{d}}\right)V \tag{2}$$

In the formula, $W^Q$, $W^K$, and $W^V$ are the three weight matrices. After random initialization, three vectors, $Q$, $K$, and $V$, are generated according to Equation (1), and then the result $A(Q, K, V)$ weighted ion mechanism is calculated according to Equation (2). The result contains the information via the attention of all of the input data.

The ProbSparse Self-Attention proposes to calculate the sparsity measurement of each query using KL divergence:

$$M(q_i, K) = Ln \sum_{j=1}^{L_K} \exp\left(\frac{q_i k_j^T}{\sqrt{d}}\right) - \frac{1}{L_K} \sum_{j=1}^{L_K} \frac{q_i k_j^T}{\sqrt{d}} \tag{3}$$

Based on the calculated sparsity metric, each key focuses on only u main queries to achieve probsparse self-attention:

$$A(Q, K, V) = softmax\left(\frac{\overline{Q}K^T}{\sqrt{d}}\right)V \tag{4}$$

In the formula, $\overline{Q}$ is a sparse matrix with the same size as $q$, and it only contains top-u queries under sparse metric $M(q, K)$.

### 2.4. Temporal Convolutional Network (TCN) Module

TCN is a variant of a convolutional neural network for processing sequence modeling tasks. It combines RNN and CNN architectures. TCN performs better than standard recursive networks on different tasks and data sets, and it demonstrates more long-term and efficient memory. The main component of the TCN network is Dilate Causal Conv. Other components are similar to the Feedforward module, which plays a role in deepening the linear features.

### 2.5. Problem Definition

The present study abstracts the photovoltaic power prediction problem as a multistep time series prediction problem, which can be defined as a data series with an input of $I \times n$ and an output of $O \times 1$, where $I$ is the length of the input data, and $O$ is the length of the output data. For example, under a 15-min sampling frequency, if the historical data of photovoltaic power in the past 30 days are used to predict the photovoltaic power data in the future 24 h, the $I$ length is 2880, and the $O$ length is 96.

## 3. Methodology

### 3.1. Transformer Based TCNformer Solution

For the time series features of photovoltaic power data, this paper proposed a TCN-former prediction model. The structure of the model is shown in Figure 3. Based on the traditional Transformer architecture, the TCNformer model mainly includes four modules: a variable selection (VS) module, an long- and short-time series feature extraction (LSTFE) module, an Encoder, and a Decoder.

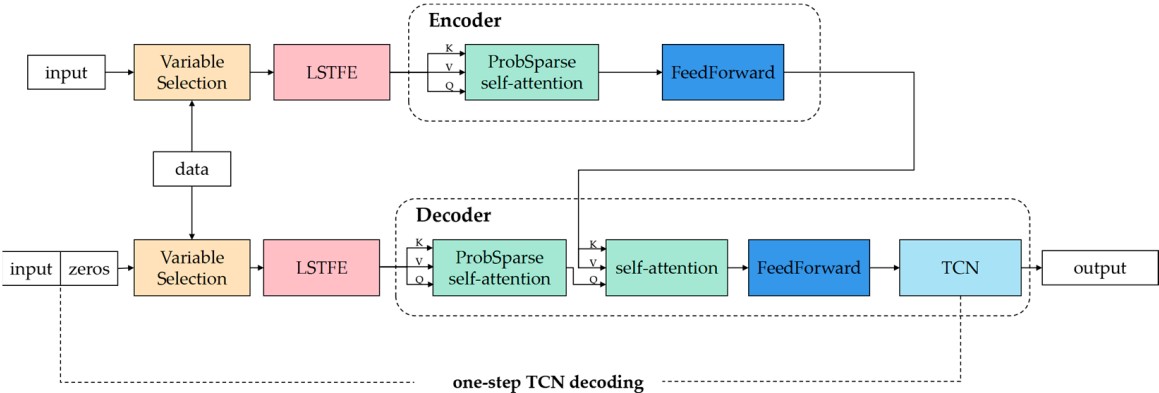

**Figure 3.** The structure of TCNformer model.

The overall TCNformer network design follows the traditional Transformer structure, in which the Encoder module and the Decoder module are designed with a multilayer structure.

### 3.2. Variable Selection (VS) Module

Combined with the information shown in Figures 1 and 2, the historical data of photovoltaic power not only have timing features in the short term, but they also have certain timing features over the long term. Considering the length of the long-term cycle (as shown in Figure 2, the cycle is close to 365 days) and the subsequent optimization problems, it is difficult for the traditional model to capture these timing features at the same time. So, we designed a VS module to divide the input sequence into three dimensions through preliminary analysis and selection of the historical data. Then, the results from the VS module are transferred to the LSTFE module for feature fusion.

$$d_l, d_s, d_t = VariableSelection(data, input) \tag{5}$$

In the formula, $data \in \mathbb{R}^{I \times n}, d_l \in \mathbb{R}^{I_l \times n_l}, d_s \in \mathbb{R}^{I_s \times n_s}, d_t \in \mathbb{R}^{I \times n_t}$ respectively represent preprocessed raw data, month-level time series data, week-level time series data, and day-level time series data. $n$, $n_l$, $n_s$ and $n_t$ respectively represent the number of influencing factors. $VariableSelection(\cdot)$ represents the VS module, and the specific calculation method is as follows.

Photovoltaic power data often show strong time series features. Although the volatility is strong, they still have a certain periodicity over a longer time range. In this paper, the Fourier transform decomposition curve of photovoltaic power data and its influencing factor data are selected for periodicity analysis [32] in order to obtain the fluctuation periods of different periodic curves and to provide a certain degree of reference for the analysis of photovoltaic power prediction. The formula of the Fourier transform is as follows:

$$X(k) = \sum_{n=0}^{N-1} x(n) W_N^{nK}, k = 0, 1, \ldots, N-1 \tag{6}$$

$$x(n) = (1/N) \sum_{n=0}^{N-1} X(k) W_N^{-nK}, k = 0, \ldots, N-1 \tag{7}$$

$$W_N^{nK} = e^{-j(2\pi/N)kn} \tag{8}$$

$X(k)$ represents the Fourier series, $x(n)$ represents the Fourier coefficient, $W_N^{nK}$ represents the complex function, $k$ represents the $x$ coordinate in the frequency domain, and $N$ represents the period.

Photovoltaic power is correlated with a large number of weather factors, especially the strong correlation between solar radiation intensity and photovoltaic power. In this study,

the Pearson correlation coefficient was selected for correlation analysis, and the calculation formula is as follows:

$$P_{x,y} = \frac{cov(x,y)}{\sigma_x \sigma_y} = \frac{E[(x_i - \overline{x})(y_i - \overline{y})]}{\sigma_x \sigma_y} \tag{9}$$

The VS module processes the month-level time series data, week-level time series data, and day-level time series data according to the analytical results of correlation and periodicity.

### 3.3. Long- and Short-Time Series Feature Extraction (LSTFE) Module

In this study, we designed an LSTFE module, and we used it to extract time series features from each time scale. The structure of the LSTFE module is shown in Figure 4. The LSTFE mainly includes the LSTM unit, the SkipGRU unit, and the CycleEmbed unit.

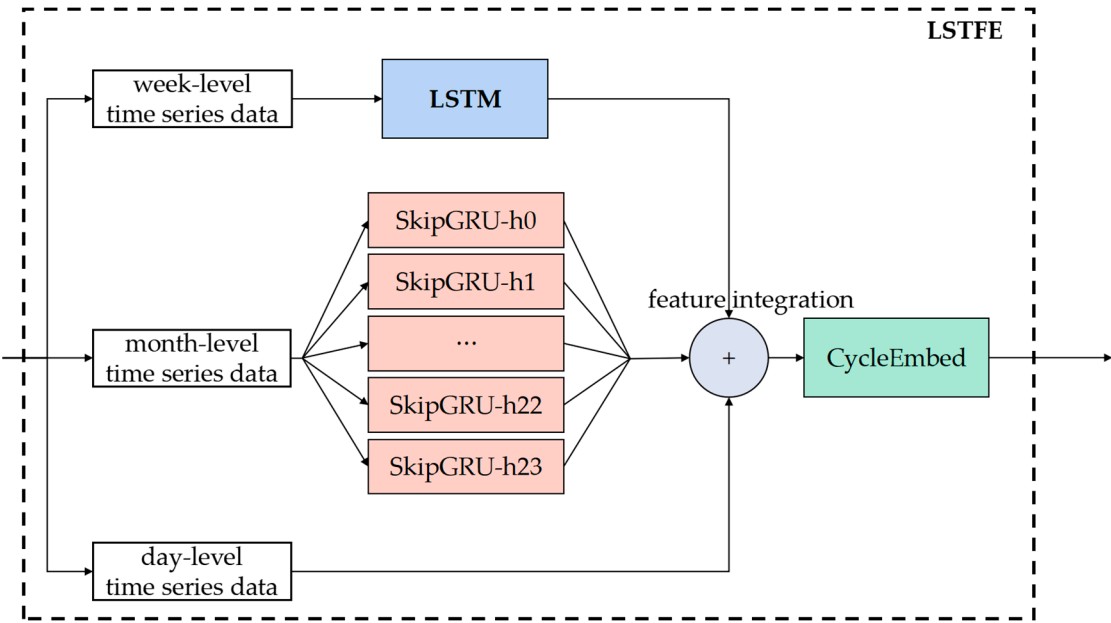

**Figure 4.** The structure of LSTFE module.

We transferred the week-level time-series-related data and the month-level time-series-related data to the LSTM network and the SkipGRU network in the LSTFE module for prediction. The prediction results of the LSTM network made full use of the short-term time series features, while the SkipGRU network made full use of the long-term time series features:

$$f_l = LSTM(d_l) \tag{10}$$

$$f_s = SkipGRU(d_s) \tag{11}$$

$$X = Integration(d_t, f_l, f_s) \tag{12}$$

$$X_{en}^0 = CycleEmbed(X) \tag{13}$$

In Formulas (10) and (11), $f_l \in \mathbb{R}^I$ and $f_s \in \mathbb{R}^I$ represent the month-level time series feature extraction results and the week-level time series feature extraction results in the LSTFE module, respectively. Using the excellent feature extraction capabilities of the LSTM and the SkipGRU, the extracted feature results were transformed into the input length $I$ of the Encoder module.

Using the LSTM and the SkipGRU, the time series features at weekly and monthly levels were extracted, but we were left wondering how to extract the time series features at an annual level? To solve this problem, we designed the CycleEmbed module.

The structure of the CycleEmbed unit is shown in Figure 5, including data projection, position coding, cycle coding, and timing coding.

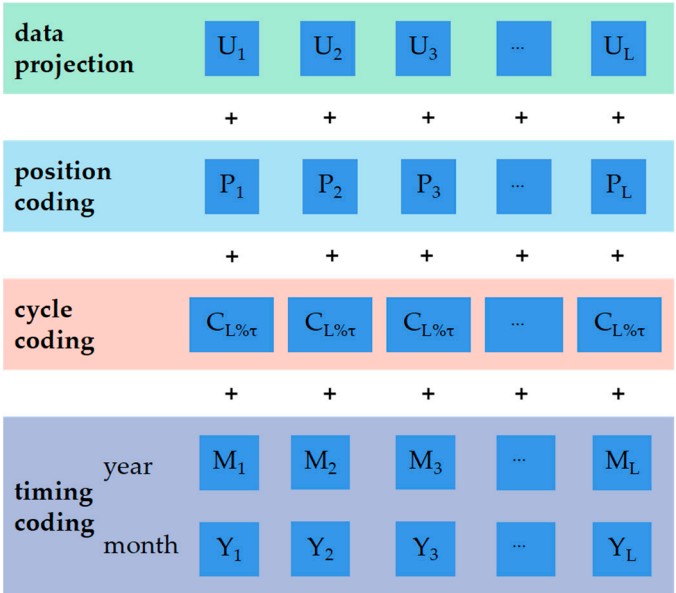

**Figure 5.** The structure of the CycleEmbed module.

Data projection is based on the results of correlation and periodic analysis, mapping the output data to the vector of dimension, and aligning the dimensions. The alignment tool is a one-dimensional convolution filter.

The position coding is calculated in the same way as in Transformer:

$$P(pos, 2j) = \sin\left(\frac{pos}{(2L_x)^{\frac{2j}{d_{model}}}}\right) \tag{14}$$

$$P(pos, 2j+1) = \cos\left(\frac{pos}{(2L_x)^{\frac{2j}{d_{model}}}}\right). \tag{15}$$

In Formulas (14) and (15), $j \in \left\{1, \ldots, \left|\frac{d_{model}}{2}\right|\right\}$, $L_x$ is the input sequence length, and $d_{model}$ is the Encoder input dimension.

Cycle coding is divided according to the results of periodic analysis and calculation. $\tau$ is the number of cycle data steps, which is determined by the results of periodic analysis $T$ and the granularity of the data sampling time $g$; that is, $\tau = T/g$. Then, the cycle information of the input data is coded according to the results of $\tau$; that is, there are $\tau$ results in cycle coding, $C_i = C_{i\%\tau}$.

Timing coding is used to add the month and year to the coding to extract the longer time series features. In this way, the annual time series features of the data are introduced into the codec along with the embedding operation.

Combining the results of the four parts, the output result of the final period embedding module is the input of the Encoder:

$$CycleEmbed_t[i] = U_i + P(L_x(t-1) + i) + C_i + M_i + Y_i \tag{16}$$

### 3.4. Encoder

The input of the Encoder is the output of the LSTFE module. The structure of the Encoder is a multilayer network structure. Each layer of the Encoder is mainly composed of a sparse attention unit and a composition unit.

$$S_{en}^{l,1} = ProbSelfAttention\left(X_{en}^{l-1}\right) \tag{17}$$

$$S_{en}^{l,2} = FeedForward\left(S_{en}^{l,1}\right) \tag{18}$$

$$X_{en}^{l} = S_{en}^{l,2} \tag{19}$$

In Formula (17), $S_{en}^{l,1} \in \mathbb{R}^{I \times d_{model}}$ is the calculation result of the sparse attention mechanism in the Layer l Encoder module, $S_{en}^{l,2} \in \mathbb{R}^{I \times d_{model}}$ is the calculation result of the Feedforward layer in the Layer l Encoder module, and $FeedForward(\cdot)$ is an important part of the traditional Transformer network structure which is used to deepen the linear representation and better extract the features. The Feedforward structure used in this paper is shown in Figure 6. ProbSelfAttention$(\cdot)$ is the sparse attention mechanism in the Informer model [24].

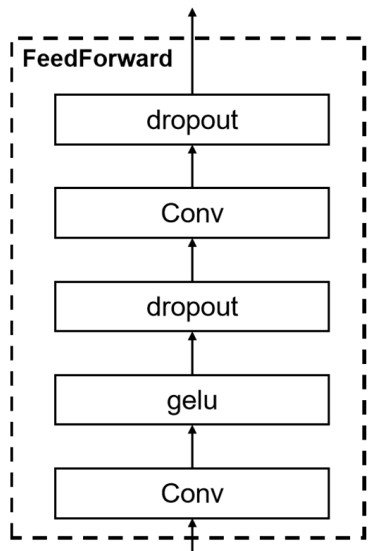

**Figure 6.** The structure of the Feedforward layer.

### 3.5. Decoder

In the Transformer model, the Encoder can be calculated in parallel, but the Decoder needs to decode step by step. As with the LSTM model, error accumulation will occur. This study introduced a one-step TCN decoding operation:

$$X_0 = Zeros[O, d] \tag{20}$$

$$X_{des} = concat(X, X_0) \tag{21}$$

$$X_{de}^0 = CycleEmbed(X_{des}) \tag{22}$$

In Formula (20), $X_0$ is the result of the zero-filling operation. One-step decoding divides the Decoder's input into two parts through a zero-filling operation. The first $I$ datum is a known sequence, the last $O$ datum is a sequence to be predicted, and $X_{de}^0 \in \mathbb{R}^{(I+O) \times d_{model}}$ is the Decoder's input data. At this time, part of the time information of the data to be predicted is also transmitted to the Decoder through the period embedding module for

prediction. The prediction process of the Decoder is similar to that of the Encoder, but it has a more of a self-attention layer than does the Encoder.

$$S_{de}^{l,1} = ProbSelfAttention\left(X_{de}^{l-1}\right) \tag{23}$$

$$S_{de}^{l,2} = SelfAttention\left(S_{de}^{l,1}, X_{en}^{N}\right) \tag{24}$$

$$S_{de}^{l,3} = FeedForward\left(S_{de}^{l,2}\right) \tag{25}$$

$$X_{de}^{l} = S_{de}^{l,3} \tag{26}$$

In Formulas (23)–(25), $S_{de}^{l,1} \in \mathbb{R}^{(I+O) \times d_{model}}$ is the calculation result of the sparse attention mechanism in the Layer l Decoder module, $S_{de}^{l,2} \in \mathbb{R}^{(I+O) \times d_{model}}$ is the result of matching the sparse attention mechanism in the Layer l Decoder module with the feature map obtained in the Encoder, and $S_{de}^{l,3} \in \mathbb{R}^{(I+O) \times d_{model}}$ is the calculation result of the Feedforward layer in the Layer l Decoder module. The calculation method of FeedForward$(\cdot)$ and ProbSelfAttention$(\cdot)$ is the same as above. SelfAttention$(\cdot)$ is the self-attention mechanism. (See Section 2.3 for the calculation method.)

$$X_{pred} = TCN\left(X_{de}^{M}\right) \tag{27}$$

$X_{pred} \in \mathbb{R}^{O \times d_{model}}$ is the final prediction result of TCNfomer, which uses TCN to make generative predictions. The TCN structure used in this paper is shown in Figure 7.

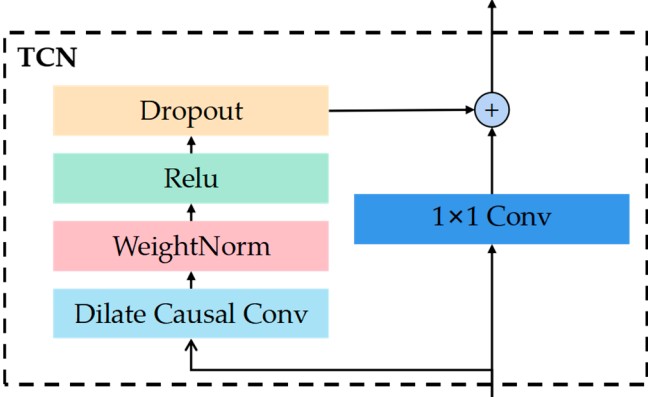

**Figure 7.** The structure of the TCN.

## 4. Experiment

### 4.1. Experimental Design

#### 4.1.1. Data Preparation

The data set included an open-source data set of photovoltaic power conducted on a solar farm in Australia [33] from 2015 to 2016. The time interval is 15 min, there are 96 data points every day, and there are 70,176 samples in total. Each sample contains 13 data, including a time stamp, received active energy, average value at the current stage, active power, performance ratio, wind speed, weather temperature in Celsius, weather relative humidity, global horizontal radiation, diffuse horizontal radiation, wind direction, daily rainfall, global tilt of radiation, and diffuse tilt of radiation. The test set used data from the last 2 months of 2016.

All data for two years are shown in Figure 8. The x-axis is the number of days, the y-axis is 96 time points per day (sampling granularity is 15 min, and 24-h data processing includes 96 event points), and the z-axis is the photovoltaic power data.

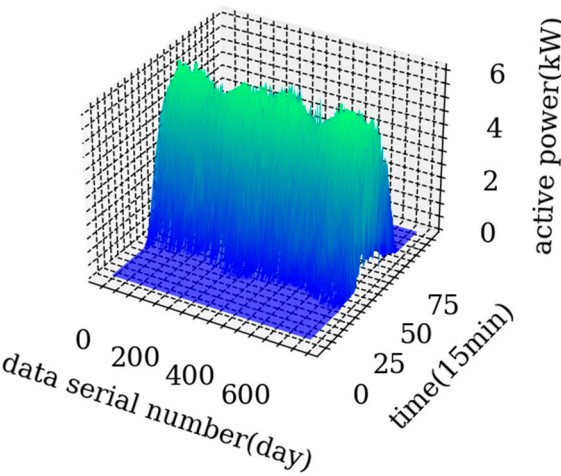

**Figure 8.** Historical data set of photovoltaic power.

4.1.2. Data Preprocessing

Since the dimensions between variables are not identical, linear normalization is required for prediction, and the conversion function is:

$$x_{norm} = \frac{x_i - min(x_i)}{max(x_i) - min(x_i)} \tag{28}$$

In the formula, $x_{norm}$ is the preprocessing result of the data after linear normalization; $x_i$ is the variable input value to be normalized; $max(x_i)$ is the maximum value of the variable in the original dataset $x_i$; and $min(x_i)$ is the minimum value of the variable in the original data set $x_i$.

4.1.3. Evaluation Index

In order to verify the prediction accuracy of the model, the root mean square error (MSE), the average absolute error (MAE) and mean absolute percentage error (MAPE) were used as the evaluation indicators of the model performance. The specific calculation formula is:

$$MSE = \frac{1}{N} \sum_{i=1}^{n} \left( X_i - \hat{X}_i \right)^2 \tag{29}$$

$$MAE = \frac{1}{N} \sum_{i=i}^{N} \left| X_i - \hat{X}_i \right| \tag{30}$$

$$MAPE = \frac{100\%}{N} \sum_{i=i}^{N} \left| \frac{X_i - \hat{X}_i}{X_i} \right| \tag{31}$$

$X_i$ is the actual output value of the $i$ th data point of the test set; $\hat{X}_i$ is the output prediction value of the $i$ th data point; and $N$ is the total number of samples in the test set.

4.1.4. Experimental Environment and Parameter Setting

The experimental environment used an Intel i7-9700 K processor and an NVIDIA GeForce RTX 3080Ti graphics card, and the algorithm model used Python 3.8 as the programming language. The model-related network was built based on the open-source machine learning framework PyTorch. The Python libraries directly used in the experiment included: pandas, numpy, matplotlib, torch, math, and time. In this study, the random search method was used to determine the final super parameter settings. The final super parameter settings are shown in Table 1.

**Table 1.** Model parameter setting.

| Parameter | Value |
|---|---|
| LSTM hidden layers | 2 |
| SkipGRU hidden layers | 2 |
| Encoder layers | 2 |
| LSTM hidden unit | 64 |
| SkipGRU hidden unit | 64 |
| Decoder layers | 1 |
| $d_{model}$ | 512 |
| batch-size | 8 |
| learn_rate | 0.0001 |
| epochs | 2000 |

### 4.2. Variable Selection Results and Discussion

The VS module in the long- and short-sequence correction network includes correlation analysis and periodicity analysis. The results of the correlation analysis on photovoltaic power are shown in Table 2.

**Table 2.** Results of correlation analysis.

| Variable | $p$ |
|---|---|
| Wind speed | 0.2096 |
| Temperature | 0.4246 |
| Humidity | −0.4072 |
| Direct radiation | 0.9690 |
| Scattered radiation | 0.5183 |
| Wind direction | −0.0444 |
| Rainfall | −0.0244 |

It can be seen from Table 2 that photovoltaic power is positively correlated with direct radiation intensity, scattered radiation intensity, temperature, and wind speed, while it is negatively correlated with humidity, wind direction, and rainfall. According to their numerical values, the data were filtered by 0.1. It can be seen that the correlation between direct radiation intensity and photovoltaic power is the largest, while variables such as scattered radiation intensity, temperature, humidity, and wind speed have a certain correlation with photovoltaic power, which show that these influencing factors have a certain degree of impact on the photovoltaic power, and the impact decreases in turn. Although wind direction and rainfall are negatively related to the photovoltaic power, the value is too small to impact the output.

It can be seen from Table 3 that the cycle of photovoltaic power, humidity, direct radiation intensity, and scattered radiation intensity is 24.03 h, approximately 1 day, and the cycle of the wind speed, wind direction, and rainfall is 0.17 h, which can be regarded as a periodicity. The temperature cycle is 8760 h; that is, the temperature cycle conforms to the changes in the four seasons, and the above results basically conform to the natural logic.

**Table 3.** Results of periodicity analysis.

| Variable | Cycle |
|---|---|
| Active power | 24.03 |
| Wind speed | 0.17 |
| Temperature | 8760 |
| Humidity | 24.03 |
| Direct radiation | 24.03 |
| Scattered radiation | 24.03 |
| Wind direction | 0.17 |
| Rainfall | 0.17 |

By correlation analysis, five influencing factors should be selected, including direct radiation, scattered radiation, temperature, humidity, and wind speed. Three influencing factors, namely, direct radiation, scattered radiation, and humidity, were screened through periodic analysis. Finally, the time series related variables of photovoltaic power were screened through the VS module, those being direct radiation, scattered radiation, and humidity.

### 4.3. Prediction Results of Different Prediction Steps

In order to explore the prediction performance of each model under different prediction steps, this study selected LSTM, SkipGRU, Transformer, and Informer to compare with TCNformer.

The results are shown in Table 4. It can be seen from the results that, when the number of prediction steps is 1, the MSE errors of the five models have little difference. With the increase in the number of the prediction steps, the LSTM model demonstrated the largest error growth rate, and the error accumulation is obvious. Informer and TCNformer use the generative prediction method, so the error was relatively stable, and the error accumulation was low. The TCNformer model proposed in this paper not only had a low level of error accumulation, but it also had the lowest MSE error. In order to more intuitively observe the error accumulation in the models, the prediction results were visualized, as shown in Figure 9.

**Table 4.** Prediction Accuracy (MSE) Results of periodicity analysis.

|    | LSTM   | SkipGRU | Transformer | Informer | TCNformer |
|----|--------|---------|-------------|----------|-----------|
| 1  | 0.1409 | 0.1272  | 0.2188      | 0.2273   | 0.0395    |
| 8  | 0.6859 | 0.3554  | 0.3135      | 0.3623   | 0.1080    |
| 16 | 0.9509 | 0.3852  | 0.3411      | 0.3644   | 0.1149    |
| 24 | 1.1062 | 0.4077  | 0.4562      | 0.5756   | 0.1322    |
| 32 | 1.3072 | 0.3501  | 0.6278      | 0.6356   | 0.1232    |
| 40 | 1.2668 | 0.4535  | 0.7122      | 0.5967   | 0.1300    |
| 48 | 1.4285 | 0.4546  | 0.6979      | 0.5571   | 0.1399    |
| 56 | 1.3687 | 0.5232  | 0.9031      | 0.6691   | 0.1234    |
| 64 | 1.4027 | 0.7204  | 0.8710      | 0.6438   | 0.1258    |
| 72 | 1.2783 | 0.7330  | 0.9645      | 0.7385   | 0.1377    |
| 80 | 1.3385 | 0.9499  | 0.8236      | 0.6510   | 0.1360    |
| 88 | 1.3601 | 1.1452  | 1.1139      | 0.7544   | 0.1382    |
| 96 | 1.4285 | 1.1624  | 1.1186      | 0.7455   | 0.1349    |

### 4.4. Prediction Performance of Different Models

In this experiment, each model was trained five times in a 24-h (96 prediction steps) scenario, and the average value was taken. The final test set prediction results are shown in Table 5.

**Table 5.** The 24-h scenario prediction results.

|             | MSE    | MAE    | MAPE     | Train Time (s) | Run Time (ms) |
|-------------|--------|--------|----------|----------------|---------------|
| LSTM        | 1.4285 | 0.7536 | 131.6989 | 139.7855       | 0.5945        |
| SkipGRU     | 1.1624 | 0.4303 | 64.8742  | 88.7615        | 0.4832        |
| Transformer | 1.1186 | 0.5385 | 3.7577   | 76.1636        | 2.2314        |
| Informer    | 0.7455 | 0.3779 | 2.9381   | 16.9919        | 1.8821        |
| TCNformer   | 0.1349 | 0.1888 | 2.4987   | 153.4314       | 1.2910        |

As shown in Table 5, the TCNformer performs best according to the three indicators of the MSE, MAE, and MAPE. Compared with the time series prediction model Informer, the MSE, MAE, and MAPE decreased by 81.90%, 50.03%, and 14.98%, respectively. The training time (153.43 s) and running time (1.29 ms) of TCNformer are relatively long, but

considering the 15-min sampling granularity and 24-h prediction scenario, the training time and running time do not affect the practical application of TCNformer.

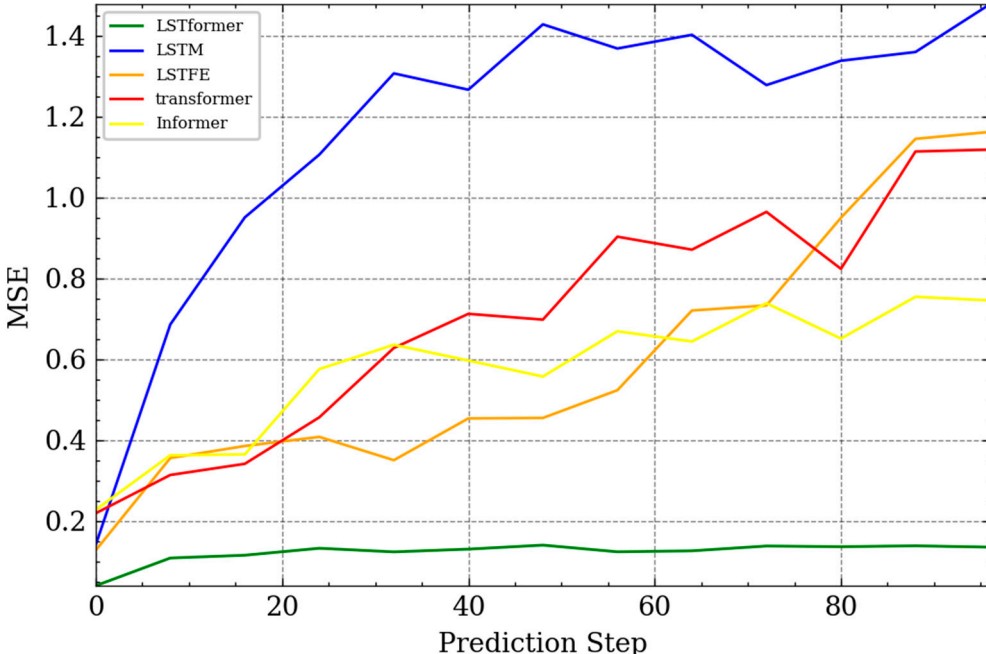

**Figure 9.** Prediction performance of the different numbers of prediction steps for each model.

As shown in Figure 10, this we visualized the prediction results of TCNformer using the test data set. The prediction results shown in the figure are 30 sets of 24-h prediction results, with little deviation when compared with the real data. It can be seen that TCNformer has a high level of accuracy and low number of errors.

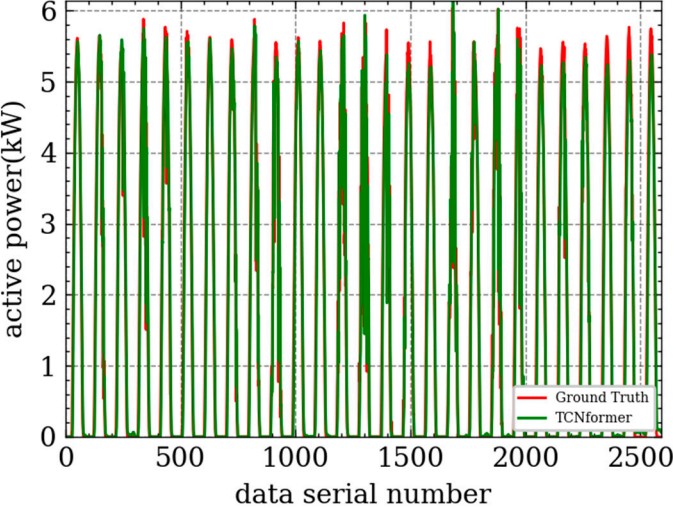

**Figure 10.** Results of the TCNformer model.

### 4.5. Error Analysis

Because the prediction of TCNformer model is a time series, we did not calculate the standard error for multiple series. Instead, error analysis was carried out through the MSE of prediction and ground truth. Figure 11 shows the standard error diagram. The error bar in the diagram represents the standard error. Table 6 shows the mean value, standard deviation (SD), and standard error (SE) of the error under different sample numbers.

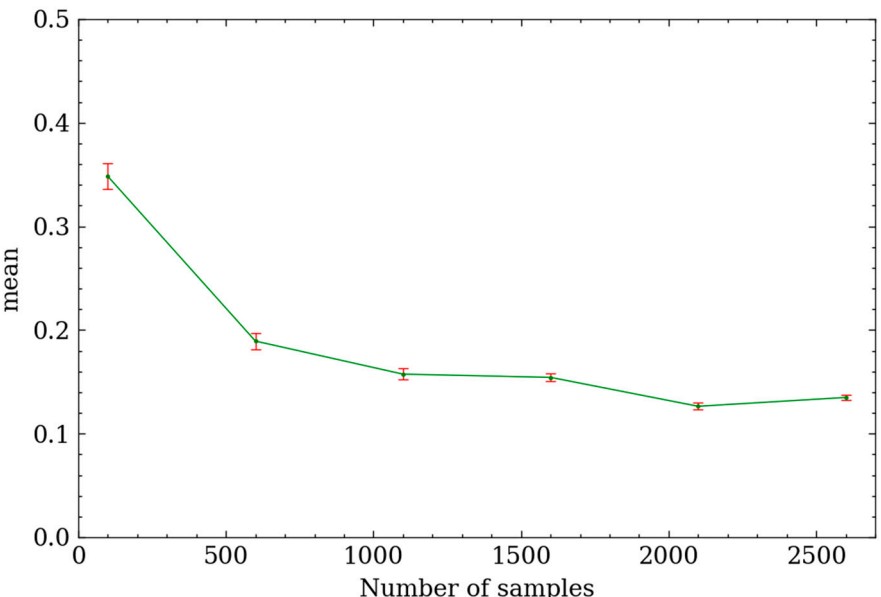

**Figure 11.** The standard error diagram of the MSE.

**Table 6.** Results of error analysis.

| Sample Numbers | Mean | SD | SE |
|---|---|---|---|
| 100 | 0.3482 | 0.1253 | 0.0125 |
| 600 | 0.1893 | 0.1977 | 0.0081 |
| 1100 | 0.1574 | 0.1798 | 0.0054 |
| 1600 | 0.1543 | 0.1584 | 0.0040 |
| 2100 | 0.1264 | 0.1507 | 0.0033 |
| 2600 | 0.1349 | 0.1394 | 0.0027 |

As shown in Table 6 and Figure 11, with the increase in the number of samples, the standard deviation and standard error gradually decreased, and the average value was closer to the average value of the overall sample. Therefore, the prediction result of the TCNformer model has a relatively stable level of error and a high level of reliability.

### 4.6. Ablation Experiment

In order to verify the effectiveness of each optimization module of the TCNformer model, we conducted ablation experiments, and we removed three innovative modules from the TCNformer model for comparative experiments, that is, we set them separately:

Experiment 1: Removal of the VS module.

Experiment 2: Removal of the long- and short-time series feature extraction module.

Experiment 3: Removal of the seq2seq structure, and use of the VS module + long- and short-time series feature extraction module + full connection network.

Experiment 4: Removal of one-step TCN decoding.

Experiment 5: Use of the complete TCNformer model.

As shown in Table 7, the three innovations proposed in this paper are a VS module, an LSTFE module, and the seq2seq generative model structure combined with Informer and Transformer. No matter which module was removed, the error of the model was increased. When the seq2seq model structure was not used, the error was the largest, and the VS module had the smallest impact on the overall model, but it still caused a decline in accuracy. From these data, it can be concluded that the TCNformer model proposed in this paper is effective, and its innovative modules are useful.

**Table 7.** Results of ablation experiment.

|  | MSE | MAE | MAPE |
|---|---|---|---|
| Experiment 1 | 0.4480 | 0.8385 | 3.5746 |
| Experiment 2 | 0.6997 | 0.5339 | 3.4223 |
| Experiment 3 | 0.8610 | 0.5288 | 4.2124 |
| Experiment 4 | 0.3695 | 0.1949 | 2.7078 |
| Experiment 5 | 0.1349 | 0.1888 | 2.4987 |

## 5. Conclusions

In this paper, a TCNformer model was proposed for photovoltaic power prediction, and we can draw the following three conclusions based on the experiment results:

1. The TCNformer model adopts the Transformer structure and introduces the sparse attention mechanism into the Informer model. The experimental results show that the photovoltaic output prediction accuracy is improved effectively.
2. The VS module, LSTFE module, and one-step TCN decoding extract more efficiently the impact of multiple time series features and other weather factors on photovoltaic power by classifying the data based on the time series, periodicity, and correlation.
3. Compared with the LSTM model and the Transformer series model, the TCNformer model has a higher level of accuracy in multistep prediction, but there is still room for optimization when the prediction range is further enlarged. In the follow-up study, we will focus on ways to solve the multistep prediction problem with a further increase in the time dimension.

**Author Contributions:** Methodology, S.L.; Software, S.L.; Validation, S.L.; Investigation, J.M.; Writing—original draft, S.L.; Writing—review & editing, D.N. All authors have read and agreed to the published version of the manuscript.

**Funding:** This research was funded by Research and development of large dispatching level data acquisition and monitoring control system grant number E212641B01 and AI assisted optimization of hybrid energy system and techno-enviro-economic analysis of green hydrogen supply chain grant number PT19797.

**Institutional Review Board Statement:** Not applicable.

**Informed Consent Statement:** Not applicable.

**Data Availability Statement:** Not applicable.

**Conflicts of Interest:** The authors declare no conflict of interest.

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
