# Peer review of "TCNformer Model for Photovoltaic Power Prediction"

_applsci, doi:10.3390/app13042593_

Round 1

Reviewer 1 Report

*The study should be reinforced with new current references.

*It is seen that this study was done very comprehensively and gave useful results. However, the results also need to be extended based on percentile % power estimates and efficiency.

Reviewer 2 Report

Shipeng Liu et al., “TCNformer model for photovoltaic power prediction”, demonstrated this article proposes a seq2seq prediction model TCNformer, which outperform the other SOTA algorithms by introducing a Variable Selection (VS), a long and short-term time series feature extraction (LSTFE) and one-step Temporal Convolutional Network (TCN) decoding. However, before any decision is made on its publication, mandatory revision is required in order to clarify some points and increase its attractiveness to the general public journal applied science. See comments below.

The introductory part (Introduction) needs to be improved. I would advise to pay attention to photovoltaic cells in order to increase the relevance of the photovoltaic industry. So the paper gives examples of the rapid growth of photovoltaic power, but I would like to see information about the maximum efficiency and about the stability problems of solar panels. Please pay attention to the following works (https://doi.org/10.1007/s00339-023-06428-0, https://doi.org/10.3390/cryst12050699, https://doi.org/10.3390/ma15228151, https://doi.org/10.3390/molecules28031288).

 I propose to reflect the standard error of the model (TCNformer) in the Conclusion.

I also ask you to improve the figures (the size of the text in the figures does not match the size of the text in the article).

Reviewer 3 Report

The authors present a prediction model that aims to improve the performance of long-term forecasting of photovoltaic power prediction. 

The paper is clear, fits the scope of the journal and is well structured.

Although the paper content has good archival value and is valuable to practicing engineers, some aspects need improvement.

I present below some comments and observations:

In the abstract, I assume that the acronyms SOTA, MAE and MSE correspond to Self-Organizing Tree Algorithm, Mean Average Error and Mean Squared Error, respectively.

The resolution of figures 1 and 8 should be improved.

Regarding Fig. 8, although the text indicates what each of the axes corresponds to, authors should introduce the captions in Fig. 8. 

What is the reason for using  Python programming language in this experiment, instead of other such as: R, Matlab, C++?

Which python libraries were used in this experiment?

On line 291, I believe you intended to write PyTorch (instead of pythoch)

Why did you use the PyTorch framework instead of the Tensorflow framework?

Why was used dataset [22] and not another dataset (as for example one of several datasets provided by Kaggle)?

Why use data from a photovoltaic station in Australia and not from a photovoltaic station in China?

Authors should make an effort to incorporate more recent references (less than 5 years old).

Regarding some formatting issues:

o The spacing between the figure captions and the text must be increased, as well as, the spacing between tables and text.

o Authors should make an effort to prevent tables from occupying more than one page (e.g. tables 1 and 3)

o Equation 4 must be placed in the center.

o In section 4.3 the table is at the beginning, without any reference to it.

o The title of section 5 should be placed on the next page.

o Apply “justify text” in conclusion.
